# Prune and distill: similar reformatting of image information along rat visual cortex and deep neural networks

**Paolo Muratore[1], Sina Tafazoli[1,2], Eugenio Piasini[1], Alessandro Laio[1,3], Davide Zoccolan[§,1]**

[1]International School for Advanced Studies (SISSA), Trieste, Italy
[2]Princeton Neuroscience Institute, Princeton University, Princeton, NJ, United States of America
[3]Abdus Salam International Centre for Theoretical Physics (ICTP), Trieste, Italy

[§]To whom correspondence should be addressed: `zoccolan@sissa.it`

## Abstract

Visual object recognition has been extensively studied in both neuroscience and computer vision. Recently, the most popular class of artificial systems for this task, deep convolutional neural networks (CNNs), has been shown to provide excellent models for its functional analogue in the brain, the ventral stream in visual cortex. This has prompted questions on what, if any, are the common principles underlying the reformatting of visual information as it flows through a CNN or the ventral stream. Here we consider some prominent statistical patterns that are known to exist in the internal representations of either CNNs or the visual cortex and look for them in the other system. We show that intrinsic dimensionality (ID) of object representations along the rat homologue of the ventral stream presents two distinct expansion-contraction phases, as previously shown for CNNs. Conversely, in CNNs, we show that training results in both distillation and active pruning (mirroring the increase in ID) of low- to middle-level image information in single units, as representations gain the ability to support invariant discrimination, in agreement with previous observations in rat visual cortex. Taken together, our findings suggest that CNNs and visual cortex share a similarly tight relationship between dimensionality expansion/reduction of object representations and reformatting of image information.

## 1 Introduction

Deep Convolutional Neural Networks (CNNs) currently stand as our best class of models of visual processing in the brain [1, 2, 3], showing success in: (1) predicting the tuning of individual neurons [4] and bold responses [5] at various stages of the ventral stream; (2) accounting for the ability of ventral stream neurons to encode a variety of object properties [6]; and (3) controlling their activity via synthetic stimuli inferred through model inversion [7, 8]. This suggests that the objective-optimization framework of deep learning offers a parsimonious explanation of the inner workings of complex, hierarchical brain circuits [9], although the latter are likely shaped by very different learning processes (e.g., unsupervised adaptation to the spatiotemporal statistics of the visual input [10, 11]. Despite this success, key differences between biological and artificial hierarchical networks exist (e.g., in sensitivity to noise or adversarial examples [12, 13]), possibly highlighting core dissimilarities in how information is processed in the two systems.

In this study, we investigated whether a similar reformatting of image information takes place along rat visual cortex and a representative CNN (VGG-16). We started from the observation that the

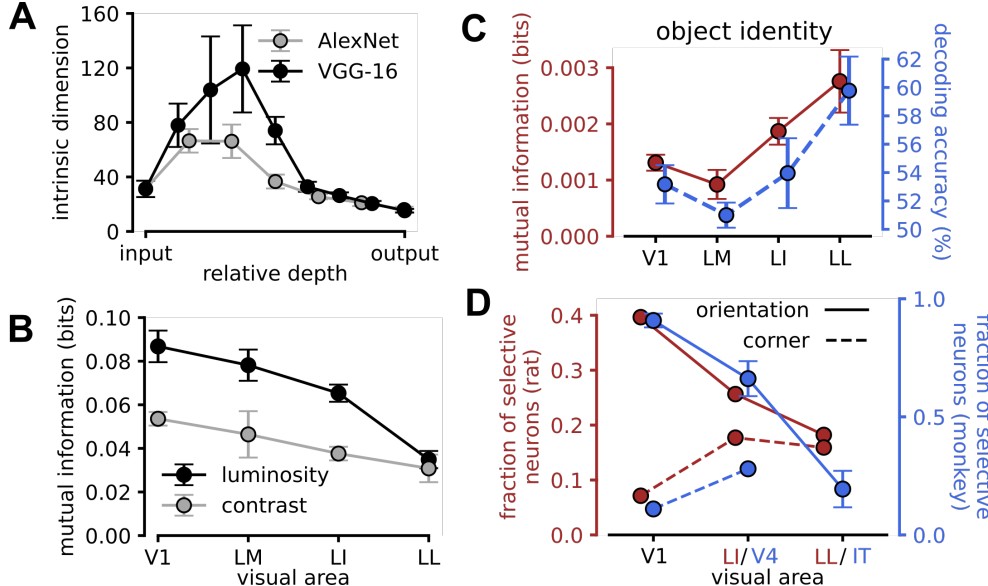

Figure 1: **Summary of previous results in CNNs and visual cortex** (**A**) Intrinsic Dimension (ID) of object representations in two deep CNNs, plotted as a function of relative depth. The error bars are standard deviations of the ID across repeated estimates (reproduced from [14]). (**B**) Median information (± SE) conveyed by single neurons recorded along the rat ventral stream (i.e., areas V1, LM, LI and LL) about luminosity and contrast of visual objects (reproduced from [15]). (**C**) Red curve: median information (± SE) conveyed by single neurons in the four areas about the identity of visual objects, each presented across a variety of distinct views. Blue curve: performance of binary linear classifiers that were trained to discriminate two visual objects (each presented across a set of different views), based on the responses evoked by the objects in the neuronal population recorded in a given area (performance was measured over a set of validation views). Error bars are SE computed over different object pairs. Both panels are reproduced from [15]. (**D**) Fraction of orientation tuned neurons (solid lines) and corner tuned neurons (dashed lines) in three different areas of the rat (red: V1, LI and LL) and monkey (blue: V1, V4 and IT) ventral stream (reproduced from [16] and [17]).

Intrinsic Dimension (ID) of object representations follows a characteristic hunchback profile across the layers of deep networks, with an initial expansion phase, followed by a strong contraction [14] (see examples in Figure 1A). This trend is so systematic across network architectures to raise the question of what information processing goals the two phases underlie. A possibility is that the initial expansion reflects the need of removing gradients of task-irrelevant, low-level features (e.g., luminosity and contrast) that are present in the visual input, while the decrease underlies a gradual reformatting of the data in pursuit of a representation better suited for the classification task [14, 18]. This idea is reminiscent of the pruning of luminosity and contrast information, and the increase of invariance of object representations that takes place along the rat homologue of the primate ventral visual pathway [15] (Figures 1B and 1C). This parallel suggests an intriguing similarity at the level of core data-reformatting processes between artificial and natural visual hierarchies. However, it is unclear if the dimensionality of object representations follows a hunchback trend in rat visual cortex, and, conversely, if luminosity and contrast information is actively discarded across CNN layers. Moreover, other (middle-level) tuning properties exist that follow characteristic trends of variation along both the monkey and rat ventral streams [16]. These are: (1) the fraction of neurons tuned for orientation, which decreases from primary visual cortex (V1) to higher-order areas (Figure 1D solid lines); and (2) the fraction of neurons tuned for multiple orientations (Figure 1D, dashed lines), a property thought to reflect the ability to encode corners [17], which instead increases from V1 to downstream areas. A few studies have reported similar trends in deep networks when probed with oriented gratings [6, 16, 19], although we still lack a general assessment of how the encoding of orientation and corner information found in natural images evolve across the layers of CNNs.

The goal of our study is to understand which of these image reformatting trends found in either CNNs or visual cortex are also a signature of information processing in the other system. Specifically, we

first analysed neuronal recordings from [15] to measure the ID of object representations across the rat ventral stream, finding also in the rat the two distinct expansion-contraction phases first described in [14] for CNNs. We then measured the information encoded by single units in VGG-16 about a variety of visual properties of increasing complexity, finding that training the network both actively distills and prunes low- to middle-level image information, in agreement with biological observations. Finally, we tracked the evolution of information about object identity at both the single-unit and population level across VGG-16 layers, exposing how such high-level information emerges sharply in late layers, again in agreement with biological findings and previous analyses of CNNs [18].

## 2 Methods

### 2.1 Analysis of neural data

To measure how the intrinsic dimension of object representations evolves across a visual cortical processing hierarchy we analyzed the dataset recorded by [15]. These data consist of extracellular neuronal responses sampled from four visual cortical areas (228 units from V1, 131 from LM, 260 from LI, and 152 from LL), while anesthetized rats were presented with a battery of 380 stimulus conditions – i.e., 38 different views (obtained by scaling, translation, rotation, etc.) of 10 visual objects. As summarized in Figure 1B-D, object representations along this pathway were found to encode stimulus information in a way that is consistent with the existence of a functional object processing hierarchy. In our study, we computed the response of each recorded unit to every stimulus as the average number of spikes fired by the neuron across repeated presentation of the stimulus within a neuron-specific spike-count window, as defined in [15]. Each stimulus condition could thus be represented by a neuronal population vector, whose components were the responses of the neurons recorded in a given area to that stimulus. The cloud of population vectors associated to the whole set of 380 object conditions formed a data manifold, whose intrinsic dimension was measured using the nonlinear estimator defined in [20] and previously applied to the analysis of CNNs in [14].

### 2.2 Dataset, network architecture and estimation of the mutual information between unit activation and image properties

We studied the behavior of the PyTorch implementation of VGG-16 [21], either randomly initialized or pre-trained on the full ImageNet dataset, with the goal of understanding how different image properties were encoded by individual units of the network as the result of training. We selected a random sub-population of 250 units from each convolutional layer (before the ReLU activation) and from the final fully-connected layers, and recorded their activations when exposed to 1500 input images taken from the ILSVRC2012 ImageNet validation dataset. Inspired by the approach applied by [15] to study rat visual cortex (see Figure 1B-D), we computed Shannon mutual information $I_i^\ell \left( X_i^\ell; Y_i^\ell \right)$ between the activation $Y_i^\ell$ of the $i$-th unit in layer $\ell$ (referred to as $u_i^\ell$ in what follows) and a given image feature $X_i^\ell$ (e.g., luminosity or contrast). The network architecture imposes for each unit $u_i^\ell$ a receptive field (RF), namely a sub-patch (denoted as $\mathsf{Image}_{\mathrm{RF}}$) of the whole image that the unit processes. As detailed in the next sections, the feature metric $X_i^\ell$ is computed by applying a specific function feat that maps $\mathsf{Image}_{\mathrm{RF}}$ to a real number (e.g., the luminosity intensity in the image patch) or a combination of real numbers (e.g., the two main orientations in the image patch), i.e., feat : $\mathsf{Image}_{\mathrm{RF}} \to \mathbb{R}$ or feat : $\mathsf{Image}_{\mathrm{RF}} \to \mathbb{R}^2$.

Given unit $u_i^\ell$, the values taken by its activation $Y_i^\ell$ over the set of input images yield a unit-specific activation distribution $p_Y(y)$ (for simplicity, we dropped the unit and layer indexes). Similarly, the values taken by the feature metric $X_i^\ell$ yield a unit-specific (i.e., RF-specific) distribution $p_X(x)$ (e.g., of luminosity intensity levels). In our experiments, both distributions were discretized into 20 equi-spaced bins. By computing the joint distribution of activation and feature values $p_{X,Y}(x,y)$, we estimated, for each unit, the mutual information between activation and feature metric:

$$I(X;Y) = \sum_{x \in \mathcal{X}, y \in \mathcal{Y}} p_{X,Y}(x,y) \log \frac{p_{X,Y}(x,y)}{p_X(x) \, p_Y(y)}. \tag{1}$$

To allow for a better comparison among the various layers and the different image features used in our analysis, the mutual information was normalized by the entropy of the distribution of the feature metric

$H(X) = -\sum_{x \in \mathcal{X}} p_X(x) \log p_X(x)$. The final estimate of the information conveyed by the units of a given layer about the feature metric was computed as $U^\ell(X|Y) = \mathbb{E}_i \left[ I_i^\ell(X_i^\ell; Y_i^\ell) / H_i^\ell(X_i^\ell) \right]$, where $\mathbb{E}_i$ is the expected value over all units $i$ of layer $\ell$. Importantly, although such unit-averaging was performed on a sub-population of $\mathcal{O}(10^2)$ units, the variability of $\bar{U}^\ell$ across independent experiment realizations (different units and stimuli) was very low, as shown by the error bars reported in Figures 3 and 5. The limited sampling bias for the mutual information was corrected with the the Panzeri-Treves method [22, 23]. Finally, we stress here how $U^\ell$, being a single-unit information estimate, is not bound by the data processing inequality and can in general express non-monotonic behaviours as a function of layer depth $\ell$.

### 2.3 Definition of the metrics to quantify visual features

In our analysis, each image patch $\mathsf{Image}_{\mathrm{RF}}$ falling within the RF of a unit was quantified by an array of four different visual properties of increasing complexity: 1) luminosity; 2) contrast; 3) orientation of the dominant edge (if any); and 4) orientations of the two dominant edges (if any), which is a proxy for the orientation and width of the dominant corner. Therefore feat $\in \{$luminosity, contrast, orientation, corner$\}$.

Luminosity can be easily defined as the average pixel-intensity in the image path: luminosity $=$ mean $(\mathsf{Image}_{\mathrm{RF}})$. Contrast quantifies the amount of luminosity variation in the patch and was computed as: contrast $=$ mean $(\mathsf{Sobel} * \mathsf{Image}_{\mathrm{RF}})$, where Sobel denotes the Sobel kernel and $*$ is the convolution operator (the Sobel transform is a standard approach to compute image gradients [24]).

The dominant orientation of in an image patch is less straightforward to quantify, because of the large variation in RF size across the layers of the network and the complexity of the natural scenes in ImageNet. At very low resolution, such as for individual units in early layers in VGG-16 (which have $3 \times 3$ RF size), no meaningful orientation can be computed. For units in late layers, which process the entire scene, multiple prominent orientations might coexist or not exist at all. More generally, image patches span a spectrum of scene orientation strength, ranging from those containing one or more sharp edges to those featuring none. To deal with such variability, we developed a two-stage, compute-and-filter approach. The orientation estimation routine is based on Fourier Analysis and defines the dominant orientation of the patch $\theta^\star$ as the angle of highest power of its Fourier spectrum (see Algorithm 1 in the Supplementary Material for a detailed pseudo-code of the pipeline). In addition, the function provides an orientation strength index $\xi \in [0, 1]$, which peaks for images containing at least one very sharp edge. Before measuring orientation information, we ranked the pool of sampled units in each layer by computing, for each unit, the average of the orientation strength index $\xi$ across the full set of 1500 input images. Out of the initial population of 250 units we only retained the 200 units with the largest average index. In addition, for each selected unit, we only considered the 500 images with the largest index $\xi$.

The corner feature was quantified as the pair of orientations of the two most prominent edges in the image patch. Specifically, the corner estimation routine applies Fourier analysis and peak-finding algorithms to identify the two dominant orientations $\theta_1^\star$ and $\theta_2^\star$ in a patch, along with a corner strength index $\zeta \in [0, 1]$ [17, 16], which is large when at least two orientations with similar power are detected in the Fourier spectrum, while it becomes negligible both for no-peak and single-peaked angular spectra (see Algorithm 2 in the Supplementary Materials for a pseudo-code description of the complete pipeline). We used this index following the same rationale as for the $\xi$ index of orientation strength, this time ranking units and input images based on their corner strength $\zeta$ and retaining a population of 200 units, each tested with a sample of 400 images.

## 3 Results

### 3.1 Intrinsic dimension of object representations along the rat ventral stream

We applied the nonlinear ID estimator Two-NN [20] to compute the intrinsic dimension of object representations in four visual cortical areas (V1, LM, LI and LL) of the rat ventral stream, as a function of the number of units included in the population vector space (Figure 2A, solid lines), up to the maximal number of units available in each area (circles). In addition, we extracted the asymptotic values of the ID (stars in Figure 2B) via power-law fits (dashed lines) to control for

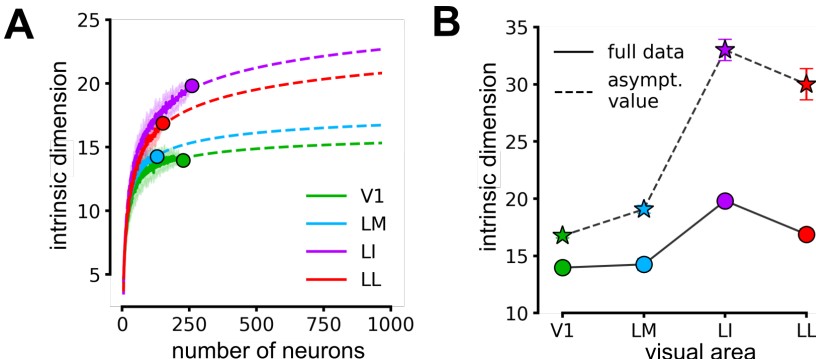

Figure 2: **Intrinsic Dimension of neural representations** (**A**) Estimation of the ID as a function of the number of neurons considered in the four visual areas (solid lines). Shadings correspond to the SD of multiple estimates with randomly sampled neuronal sub-populations, while circles mark the estimates obtained with the full populations in each area. Dashed lines are power-law fits to the data. (**B**) The ID estimates obtained for the full populations (circles) and for the asymptotic values of the fits (stars) are plotted as a function of the rank of the areas along the rat ventral stream. Error bars are the standards deviation of the values returned by the fit.

finite-size effects. At any population size considered, the ranking of the visual areas in terms of the estimated ID was remarkably stable, with V1 featuring the lowest ID, LI the highest, and LM and LL reaching intermediate values. More importantly, plotting the ID in each area as a function of its rank along the cortical processing hierarchy (Figure 2B) revealed a characteristic "hunchback" profile, with an initial expansion (from V1 to LI), followed by a contraction (from LI to LL). This trend is consistent with the one observed in deep networks (see Figure 1A) by [14], who conjectured that the initial ID expansion was due to the pruning of low-level image information (e.g., luminosity and contrast). Our result strongly supports this intuition, since the alternation of the expansion-contraction phases is now observed along an object processing pathway where such pruning has been shown to take place (see Figure 1B) [15].

## 3.2  Encoding of low- to middle-level visual features in single units of VGG-16

We now turn to the other question addressed in our study, namely investigating if the information about low-level image features is actively discarded in artificial networks in a manner that resembles the one observed in rats. Having defined a set of metrics to quantify image features of increasing complexity (see Section 2.3), we measured how much information about these features was encoded by the activation of individual units across the layers of VGG-16 (see Section 2.2 for details).

We found that information about luminosity was a monotonic decreasing function of the layer's depth, with training producing a very large luminosity information loss in the very first layer (compare blue and green curves in Figure 3A). Intuitively, this can be explained by the fact that learning spatially structured convolutional kernels will tend to produce both positive and negative weights with balanced, near-zero average, which are poorly sensitive to the mean luminosity within a unit's RF. By contrast, randomly assigned weights will often have the same sign, at least for the small kernels of the early layers, yielding activations that are proportional to the luminous energy falling within a unit's RF. This intuition was confirmed by comparing the distributions of the average weights for the $3 \times 3$ kernels of the first layer in the trained and untrained network (bar plot in the inset). The gradual monotonic decrease that was nevertheless observed in the untrained network is explained by the fact that randomly assigned weights, in case of increasingly larger kernels, will progressively tend to the zero-average condition (inset, red line).

If training produces spatially structured kernels, units in early layers should not only lose sensitivity to luminosity, but also become sensitive to image contrast. Our mutual information analysis confirmed this intuition, showing that the units of the initial layers encoded a larger amount of contrast information in the trained network, as compared to the untrained one (Figure 3B, blue vs. green curve). In addition, as a result of training, contrast information grew steadily in the early convolutional layers, reaching a peak in the third one, but then decayed sharply in the following layers, eventually

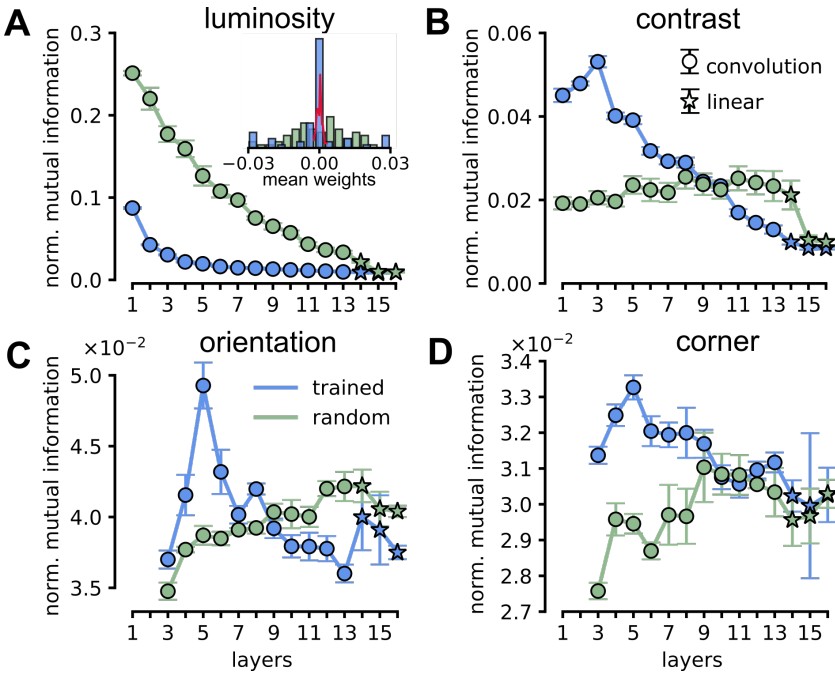

Figure 3: **Distillation and pruning of image information in VGG-16** (**A**) Mean normalized information conveyed by VGG-16 units about image luminosity for a trained (blue) and random (green) network. Error bars are standard deviations over five realizations of the experiment (independent sampling of units, images and random weights). Circles represent convolutional layers, while stars indicate fully connected layers. Inset: distribution of the average weights of the units in the first layer of the trained and random network (blue and green bars) and in the forth convolutional layer of the latter. (**B-D**) Same as in **A**, but for the information conveyed by VGG-16 units about image contrast, orientation and corners (i.e., joint orientation of two prominent edges).

attaining values that were lower than those of the untrained network. This suggests that learning representations that are useful to process and classify natural images requires to first distill contrast information in the units of early layers, followed by actively discarding such information in later processing stages. The pruning of contrast and luminosity information matches the results in rat visual cortex [15] (see Figure 1B). We note that the analysis of rat data did not reveal the initial rise of contrast information found in VGG-16, but this is unsurprising, given that the rat dataset did not contain recordings from the processing stages that precede V1, i.e., retina and thalamus, where center-surround contrast detectors first emerge in the visual system and that would correspond to VGG-16 very first layers.

We next considered visual features of increasing complexity, measuring the amount of information encoded by VGG-16 units about the dominant orientations of the image patches falling within their RFs. This analysis was applied only to units for which enough input images existed that contained, in the patch falling within the units' RFs, a sufficiently prominent oriented edge (see Section 2.3). Moreover, we excluded from the analysis the first two layers of the network, because their units have RFs that are too small for the orientation estimate to be meaningful. When visualized as a function of layer depth, orientation information in the trained network followed a hunchback profile (Figure 3C, blue curve), raising sharply and reaching a peak in the fifth convolutional layer, i.e., at a later stage than contrast information (see Figure 3B), consistently with the hierarchically higher nature of the orientation feature. Following the peak, orientation information dropped sharply in the deeper layers. As for luminosity and contrast, this trend was the result of training, as it was not observed in the randomly initialized network (green curve). And again, as for contrast, orientation information, once distilled in individual units of early layers, was then actively discarded in the following processing stages, becoming lower than for the untrained network - a finding consistent with the loss of orientation tuning found along the ventral stream [16] (see Figure 1D, solid lines). Just like for contrast, no initial rise of orientation tuning was observed along the rat ventral stream,

because no data were available from subcortical areas where orientation tuning is known to be much less prominent [25].

Finally, considering features of even greater complexity, we measured the information conveyed by VGG-16 units about the joint orientation of two dominant edges, i.e., the corner information (again, this analysis was applied only to cases where the image patch falling within a unit RF contained a sufficiently prominent corner; see Section 2.3). As for orientation, also corner information varied across the layers of the trained network according to a hunchback profile (Figure 3D, blue curve), again peaking in the fifth convolutional layer, and again being discarded in deeper processing stages, but more gradually than orientation information, reaching a sort of plateau in middle layers. Once more, this trend was not observed in the untrained network (green curve) and was instead consistent with the increase of neurons tuned for pairs of orientations found along the ventral stream [16] (see Figure 1D, dashed lines).

Importantly, all these feature information trends were largely preserved when assessed on the activations following the ReLU non-linearities (see Appendix C), and when tested on other networks of the VGG family (see Appendix D)

### 3.3 Effective pruning of low-level information requires training

One of the most intriguing findings of our experiments is that training is necessary not only to build sensitivity for low- and middle-level visual features, but also plays the complementary role of pruning this information, once it has been distilled in individual units of early layers. To better understand the extent to which information pruning is actively enforced by training, we considered a hybrid VGG-16 network constructed as follows: layers $\ell \leq \ell^\star$ shared the same weights as the fully-trained (on ImageNet) VGG-16, while weights in layers $\ell > \ell^\star$ were left randomly initialized. By letting $\ell^\star$ vary, one could visualize the effect of random transformations after a given checkpoint ($\ell^\star$) and ask whether the observed decay of feature information (Figure 3) is a direct consequence of training (active information pruning) or is merely the result of architectural constraints. We found that training played an active role in pruning luminosity information (Figure 4A), with the information profile of the fully-trained network (blue curve) being consistently lower with respect to the profiles obtained for hybrid networks with intermediate $\ell^\star$ checkpoints (green curves). The effect of training was even more striking for contrast information (Figure 4B), which, in the hybrid networks, displayed a growing trend through the random convolutional layers, before finally dropping in the second fully-connected layer. In the case of orientation (Figure 4C), results were only partially consistent with those of luminosity and contrast. Training the network only up to layer 5 (i.e., up to the peak of orientation information), still yielded a large drop of information from layer 6 onward when these layers are left randomly initialized. However, orientation information did not reach the same low values attained by the fully-trained network in the last layers: information here remained substantially higher. This indicates that a reduction of orientation information after the peak in layer 5 is achieved

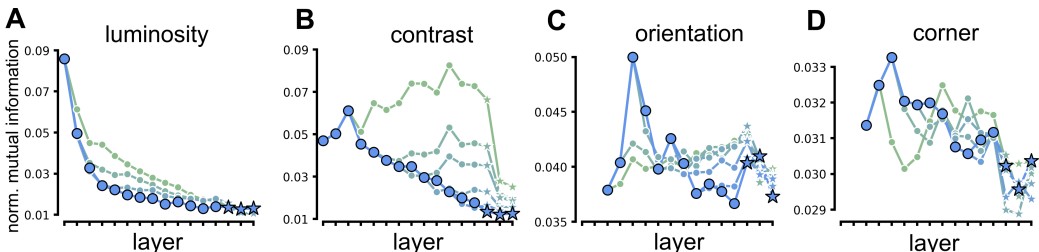

Figure 4: **Training results in active pruning of low-level image information** (**A**) Mean normalized information conveyed by VGG-16 units about image luminosity for a trained network (thick blue line; same curve as in Figure 3A) and for three additional hybrid network configurations (green lines) that have been trained only until layer $\ell^\star$ (with weights in the following layers having been left randomly initialized). The gradients of color (from green to blue) correspond to progressively larger $\ell^\star$ values, i.e., $\ell^\star \in \{1, 2, 3\}$. (**B**) Same as in **A**, but for the information conveyed by VGG-16 units about image contrast. Here the thick blue line is the same curve as in Figure 3B and $\ell^\star \in \{3, 5, 7, 9, 11\}$. (**C-D**) Same as in **A**, but for the information conveyed by VGG-16 about image orientation and corner.

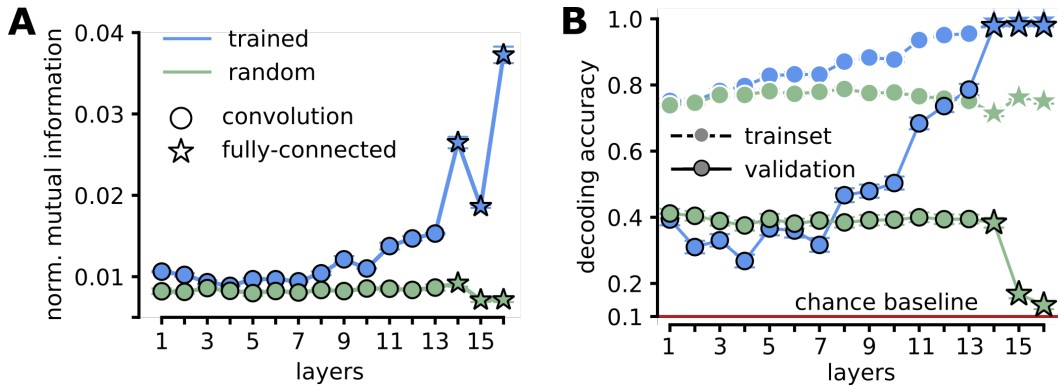

Figure 5: **Evolution of category information across VGG-16 layers** (A) Mean normalized information conveyed by VGG-16 units about image category for a trained (blue line) and a random (green line) VGG-16 network. As in Figure 3, error bars are standard deviations over five realizations of the experiment. (B) Training and validation performance (dashed and solid lines respectively) of linear SVM classifiers that were trained to predict the labels of images belonging to 10 selected Imagenet categories (250 and 50 images per category were used, respectively, for training and validation), based on the activations of a pool of 250 VGG-16 units sampled from each layer. Data points are averages (± SD) over 200 sub-populations of 100 units that were randomly sampled from each pool of 250 units.

even without training, likely because of architectural constraints (e.g., increase in receptive fields). However, further pruning of orientation information in the last layers of the network still requires training, consistently with the results found for luminosity and contrast information. A qualitatively similar behavior, albeit noisier, was found for corner information (Figure 4D).

### 3.4 Information on object identity emerges in late layers at both the single unit and population level

The VGG-16 network used in our experiments was pretrained to achieve high classification performance on Imagenet. Thus, all the information about the low- to middle-level visual features explored in our analyses must have been harvested (and then pruned) by the network in the attempt to maximize the separability of image categories in its output layer. Having reported how information about several such features peaked in early convolutional layers, we next asked how information about image category evolved across the network. Intuition suggests that it should peak in the very last layer, where readout takes place. It is however unclear how such information varies along the network depth, especially the one encoded by individual units. In the rat ventral stream, information about object category encoded by single neurons has been found to rise from low to high visual areas, with a matching increase in the ability of neuronal populations to support invariant recognition [15] (see Figure 1C). In [18], using a neighbourhood regularity metric, it was shown how representation-support for image category emerged sharply in late layers of various CNN architectures.

Here, we measured the category information encoded by VGG-16 units by using the label of the 1500 test images as the feature variable $X_i^\ell$ in Eq. (1). We found that this metric remained low and stable for about half the depth of the trained network, increasing smoothly in the last convolutional layers and then abruptly in the fully connected ones (Figure 5A, blue curve), while no trend was observed for the random network (green curve). This result resonates with that in [18], again indicating a late and sharp rise in image category information.

Next, we investigated how easily accessible was such category information encoded by single units. To this aim, we trained linear SVMs to predict the image labels based on the activity of a pool of 250 units in a layer (Supplementary Material, section B). We found a growth of decoding accuracy (Figure 5B, blue curve) that tracked the increase of category information observed, at the single unit level, in the second half of the convolutional layers (compare to Figure 5A). This suggests that, similarly to what observed along the rat ventral stream, the concentration of category information in single units plays a role in supporting the linear readout of category label at the population level.

As expected, the decoding accuracy matched the expected close-to-perfect performance in the final, fully connected layers (see Figure 5B, blue lines). Interestingly, even the random network supported above-chance decoding beyond the training domain (solid green curve), with both the trained and random configurations remaining closely tied for half of the computation depth (mirroring, again, what observed for the category information at the single unit level).

# 4 Conclusions and Discussion

The ventral visual stream [26] and deep convolutional neural networks [27] are very effective solutions to the problem of object vision. The precise extent to which these two classes of systems process visual information based on similar principles is an open question and an active field of investigation [1, 2, 3]. Here, we compared the rat ventral stream and a widely used CNN (VGG-16) by tracking how image representations are progressively reformatted across the two hierarchies. Differently from previous studies, our goal was not to use CNNs as models to predict the tuning of visual cortical neurons or the spatial structure of their RFs [4, 2, 7, 8, 28, 29, 5]. Rather, we took inspiration from recent studies showing that some key statistical properties of image representations follow very specific trends of variation across either the artificial or the cortical processing hierarchies [15, 14], and we looked for them in the other system.

In this study, we focused on the rat brain as the biological term of our comparison with CNNs, rather than on the primate brain, which certainly possesses a more advanced visual system. This is because, despite the long tradition of vision neuroscience in primates, most of the information processing trends we explored in this work have been systematically measured only in the rat [15]. This is related to the fact that, in monkeys, it is difficult to record from the whole ventral stream using the same battery of visual stimuli. Typically, no more than a pair of cortical areas are investigated in a single study (e.g., V4 and IT [30]) and object representations in V1 are often simulated rather than measured [4, 6]. By contrast, in rats it is possible to probe with the same stimuli V1 and the whole progression of lateral extrastriate areas (LM, LI and LL) that play the functional role of an object-processing pathway [31, 15, 32]. This makes it possible to analyze a cortical hierarchy that is deep enough for a meaningful comparison with a CNN. It is worth stressing that we specifically focused on the rat rather than on the mouse visual system, because evidence for the existence of an object-processing pathway is way more limited in mice (but see [33]) and, not surprisingly, deep CNNs have been found to be poor models of mouse visual cortex [29].

Our analyses yielded three main results. First, the ID of object representations across the rat ventral stream varied according to the same hunchback profile previously found by [14] in CNNs (compare Figure 2B to Figure 1A). Here it should be noted that, compared to the dramatic ID contraction observed in the final layers of CNNs, the drop found from LI to LL was much smaller. This is not surprising, because LL, despite its high rank along the rat visual cortical progression, is not the final stage of the hierarchy. In fact, the rat ventral stream possesses at least one higher-order area (TO, [31, 34]; not probed in [15]). Additionally, the deepest layers of CNNs contain representations that are highly specialized for the classification tasks they were trained on, suggesting a better match with cortical regions involved in memory and decision making (such as perirhinal, posterior parietal and prefrontal cortex) than with purely sensory areas. It is in these regions that representations may be expected to become as low dimensional as those found in CNNs' final layers [35, 36].

Our second main result is that information about low-level features (luminosity and contrast) encoded by individual units of VGG-16 was progressively pruned across the processing hierarchy, as previously found by [15] across rat visual cortical areas (compare Figure 3A-B to Figure 1B). A similar pruning was observed also in the case of orientation and corner information (Figure 3C-D), with the difference that sensitivity to these higher-order features started low and had first to be progressively distilled through processing along the first layers (a trend that, although less prominent, was also observed for contrast information). These trends are consistent with the drop of orientation tuning and the increase of tuning for multiple orientations (corners) found in both the rat and monkey ventral streams [16] (Figure 1D). Again, as for the case of the ID trend discussed above, this consistency between cortex and VGG holds in terms of global trends and not at the level of a one-to-one, area-to-layer match. In fact, while for the CNN we have access to the entire hierarchy, for the rat visual system we only have access to a subset of processing stages. In particular, the neuronal data set did not include data from both deep memory/decision areas and early processing stages (retina and thalamus). In CNNs, this would be equivalent to missing the first few convolutional layers, as well as some of

the deepest convolutional layers and the fully connected ones (i.e., rat visual areas V1, LM, LI and LL could roughly be equated to layers 5-8 in VGG-16). A tighter match between rat visual areas and CNN layers may become possible using more advanced neuronal recording technologies that allow targeting a larger numbers of visual processing stages (e.g., using Neuropixel probes [37]). Moreover, it would be interesting to extend our approach to fMRI data spanning the entire human ventral pathway [5] and estimate how much information is conveyed about the various low-to-middle level features tested in our study by the activity of individual voxels.

The feature information trends reported in our work are also consistent with the orientation tuning profiles reported across CNN layers by a other studies [6, 16, 19] and with the way the spatial structure of convolutional kernels evolves across CNN layers, where early Gabor-like kernels are replaced by filters with more complex geometries in late layers [38, 5]. Importantly, our analyses clearly show that information pruning is not a trivial byproduct of architectural constraints (e.g., RFs becoming larger as a function of layer depth), but is an active process that takes place, across the whole network, as the result of training (Figure 3 and 4). This suggests that, in hierarchical visual processing systems (biological and artificial alike), sensitivity to features that are required for the buildup of higher abstractions (e.g., contrast for edges; edges for corners; corners for shapes; etc.) might become useless or even harmful for further learning in deep layers. Thus, our work extends the findings of previous modeling studies of visual cortex using CNNs [4, 2, 7, 8, 28, 29, 5] by explicitly measuring the existence of non-monotonic trends of feature information across CNN layers and by establishing their dependence on training.

Finally, our experiments revealed that the growth of classification accuracy afforded by image representations across VGG-16 layers closely tracked the increase of category information encoded by individual units (Figure 5). This result is consistent with the tight relationship found, in the rat ventral stream, between the view-invariant object information encoded by single neurons and the power of neuronal populations to support invariant object recognition (see Figure 1C). Overall, this suggests that low/middle-level image information and higher-order categorical information trade off along visual processing hierarchies, echoing previous observations that have emphasized the role of learning in suppressing irrelevant information [39].

Taken together, these findings point to the existence of a functional relationship between dimensionality expansion/reduction of object representations and distillation/pruning of various kinds of image information, suggesting that such relationship is likely a fundamental property of both biological and artificial visual processing architectures. Further experiments are necessary to probe the generality of this conclusion across natural visual systems (i.e., recording from more visual areas and different species) and across a larger variety of artificial neural networks. In our study, we focused on VGG because it is one of the most popular CNN architectures in visual neuroscience, commonly used either as a model of the ventral stream or as a benchmark against which such models should be tested (see e.g. [5, 40, 41, 42, 43, 44, 45]). More importantly, VGG, as any other simple feedforward convolutional network, allows for a very natural definition of the receptive field of individual units. This is fundamental for our analysis, because a notion of receptive field is required for the estimation of image feature information. By contrast, more modern architectures, such as ResNets or ViTs, allow for non-trivial paths of information flow (residual connections, skip connections, or the attention mechanisms), which make the identification of receptive fields more challenging. Exploiting gradient backpropagation to the input level or other feature-visualization techniques (e.g. gradCAM [46]) may be a viable approach to overcome this issue, thus allowing extending our analysis beyond the purely feedforward convolutional framework.

## Acknowledgments and Disclosure of Funding

We thank A. Ansuini for his help on getting started with the computation of intrinsic dimension in deep nets and neuronal data. We thank D. Doimo and L. Porta for suggestions on the implementations of our analyses. We thank A. Benucci for his feedback on the interpretation of our findings.

This work was supported by a European Research Council Consolidator Grant (project no. 616803-LEARN2SEE to D.Z). We acknowledge the HPC Collaboration Agreement between SISSA and CINECA for granting access to the Marconi100 cluster.

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
