# OpenReview forum: "Prune and distill: similar reformatting of image information along rat visual cortex and deep neural networks"
_NeurIPS.cc/2022/Conference — NeurIPS 2022 Accept_

### Official Review · Reviewer_sQm2 · 2022-07-06

**Rating:** 7
**Confidence:** 4
**Soundness:** 3 good
**Presentation:** 4 excellent
**Contribution:** 2 fair

**Summary:**

The paper investigates a parallel between two processing systems: (1) a deep convolutional neural network (VGG-16), and (2) the rat visual cortex. More precisely, the paper considers previous results obtained on one system, and attempts to replicate them on the other system. These results are related to the processing of information along each system.

To investigate which information is represented in each layer of each system, the paper considers a number of metrics computed on the intermediate representations:
- intrinsic dimensionality
- mutual information with an image metric computed on the input (average luminosity, contrast, orientation, couple of orientations)
- decoding accuracy

By considering the trend of these metrics over the layers of each system, the paper demonstrates a parallel between the two systems.
The results also describe the difference in representations between a pretrained CNN and a network with random weights.

**Questions:**

1. The paper uses neurophysiological recordings in rats. The justification for this choice is that it enable recording simultaneously over many part of the visual hierarchy. Can the authors discuss the possibility of using fMRI in humans, which records simultaneously over more parts of the visual hierarchy?
2. To best match VGG-16 and the visual cortex, which subset of layers would best correspond to the four visual areas (V1-LM-LI-LL) ? Is there a way to select a subset of layers with consistent trends with the four visual areas?
3. Why is figure 4 limited to luminosity and contrast, and does not include orientation and corners as figure 3?

**Limitations:**

4. The authors briefly discuss the fact that the dataset considered is not capturing the entire visual hierarchy, thus limiting the parallel with the CNN architecture.

5. The authors have not discuss the choice of VGG-16 compared to other existing CNN architectures.

**Strengths And Weaknesses:**

**Clarity**

The paper is very clearly written in all aspects: exposition of the problem, relation with the literature, proposed method, results, and discussion.
It is well structured, and the figures are of top quality.

The results seem sufficiently described to be reproducible. Additionally, sharing the code to reproduce all the figures is appreciated (not tested).

**Originality**

The investigation is relatively original.
A large number of paper have investigated mappings between artificial and biological neural networks, particularly CNN and the visual cortex (in humans, monkeys, rats, and more), using methods such as encoding models or representation similarity analysis.

This paper follows this line of work, but it is original in that it investigates a number of manually computed metrics. These metrics have a somewhat limited complexity, but they can be computed over different layers to investigate explicitly how the information is processed in each system.


**Quality**

The experiments are sound, seem well executed, and the results are reasonable.

The paper is structured around the replication of figure 1 (ABCD) in the other system:
- figure 1A vs figure 2. The hunchback profile in the visual cortex is not as neat as in the CNN, but the results are somewhat consistent. The consistency assumes we ignore (roughly) layers 9-16.
- figure 1B vs figure 3A/B. The decreasing trends are consistent, but only if we ignore layers 1-3.
- figure 1D vs figure 3C/D. Here the trends are not as consistent as stated. In the visual cortex, the orientation representation decreases with depth, and the corner representation increases with depth. In the CNN, the orientation trend is only decreasing if we ignore layers 3-5, and the corner trend is only increasing if we ignore layers 5-16.
- figure 1C vs figure 5. The normalized mutual information mostly increases in the last layers (say 7-16).

Overall, the trend consistency between the visual cortex and the CNN is a bit overstated. The trends are only consistent if we consider a subset of layers, and this subset is not the same depending on the representation considered.


**Significance**

The results are not particularly surprising. A lot of studies have already shown a parallel between CNN and the visual cortex in mammals.
This new study adds up to this parallel, describing it with an original approach.

The pretrained-vs-random results are all unsurprising, but nicely quantified in the different experiments.

---

> ### Author Response · Authors · 2022-08-01
> **Reply to reviewer sQm2 (Part III)**
>
> ### **3. Question #1: comparison with human data**
> > *The paper uses neurophysiological recordings in rats ... Can the authors discuss the possibility of using fMRI in humans …?*
>
> We thank the reviewer for pointing this out. In general, comparison between the human visual system and CNNs can be very fruitful, as shown, for instance, by the study of Güçlü and van Gerven (2015), as highlighted by reviewer DRfj. In the case of our analysis, we do believe it would be possible to estimate how much information is conveyed about the various low-to-middle features by the activity of individual voxels.
>
>
> ### **4. Question #2: match between VGG-16 and visual cortex**
> > *To best match VGG-16 and the visual cortex, which subset of layers would best correspond to the four visual areas (V1-LM-LI-LL)? Is there a way to select a subset of layers with consistent trends with the four visual areas?*
>
> For the answer to this question, please **see point 1 of our rebuttal above.**
>
> ### **5. Question #3: missing panels form Fig. 4**
> > *Why is figure 4 limited to luminosity and contrast, and does not include orientation and corners as figure 3?*
>
> We originally carried out the analysis shown in Fig. 4, motivated by the finding that luminosity information decreased strongly also in the random network (Fig. 3A). We wanted to understand if the larger decrease observed in the trained network was due solely to the drop taking place in the first layer or if luminosity information kept being discarded along the rest of the network (Fig. 4A shows that this is indeed the case). Then, it was natural for us to extend it to another low-level feature as contrast, finding an even stronger impact of training in actively discarding contrast information (Fig. 4B). **Following the reviewer suggestions, we have now run the same analyses for the other, higher-order features (orientation and contrast).**
>
> In the case of orientation, we found a result that is only partially consistent with that of luminosity and contrast. If we train the network only up to layer 5, where there is the peak of orientation information, we still see a large drop of such information from layer 6 onward, when these layers are left untrained (i.e., randomly initialized). However, orientation information, in this hybrid network, does not reach the same low values attained by the fully-trained network in the last convolutional and fully connected layers: information here remains substantially higher. This indicates that a reduction of orientation information after the peak in layer 5 is achieved even without training, likely because of architectural constraints (e.g., receptive fields getting bigger). However, **maintaining very low orientation information in the last layers of the network is still an active process that requires training, consistently with the results found for luminosity and contrast information.** A similar behavior was also found for corner information.
>
> We will revise Fig. 4 to add these new results in the camera-ready version of our revised manuscript. The reviewer can see the figure at the following anonymous link: https://figshare.com/s/cc9d6b61cf061cfc6d86.
>
>
> ### **6. VGG vs.other CNNs**
> > *The authors have not discuss the choice of VGG-16 compared to other existing CNN architectures.*
>
> We agree that testing how our results extend to different ANN architectures is a very interesting question. This was also one of the criticisms raised by reviewer 2Y7t. Following the reply to their comment, we originally focused on VGG for two main reasons:
> 1)	It is one of the most popular and influential CNN architecture families in visual neuroscience (see the reply to 2Y7t for a selection of relevant references).
> 2)	VGG, as any other simple feedforward convolutional network, allows for a very natural definition of the receptive field of individual units, which is essential, in our analyses, for the estimation of image feature information. By contrast, with more complex architectures such as ResNets this concept would be harder to define, leading to the introduction of more hypotheses and more moving parts to the study
>
> However, motivated by the reviewers’ observations, **we extended our analyses to other networks of the VGG family** (VGG-11 and VGG-19) that, just like VGG-16, have been popular in vision neuroscience and that allow an unambiguous definition of the receptive field. In such networks, we obtain very similar results to those in VGG-16.
>
> This analysis is shown in a figure that is available at the following anonymous link: https://figshare.com/s/9a14ec02441a8f42cf08. We will add these results to the Appendix of the paper in its final version, and we will include a discussion of this point in a separate Limitations paragraph.

---

> > ### Comment · Reviewer_sQm2 · 2022-08-08
> > **Thanks for the comments**
> >
> > I thank the authors for their detailed answers. They addressed many of my concerns, so I have updated my rating accordingly.
> >
> > I still think that the significance of this work is relatively limited, but I don't think this limitation should stop this paper from being accepted for publication.

---

> ### Author Response · Authors · 2022-08-01
> **Reply to reviewer sQm2 (Part II)**
>
> The reviewer is right in pointing out such discrepancy. At the same time, it is important to emphasize that a perfect area-to-layer match between cortex and VGG-16 was hardly expected. These are systems that perform similar functions, with a globally similar architecture, but with several important differences in terms of number of processing stages and units, connectivity, local computations, learning algorithms, etc. What is important, in our opinion, is that, despite these differences, most of the information processing trends we have analyzed are qualitatively consistent in the two systems.
>
> > - *figure 1C vs figure 5. The normalized mutual information mostly increases in the last layers (say 7-16).*
>
> We believe that the different “rate” at which category information increases in the two figures is due to the different complexity of the image sets used to probe rat visual cortex and the CNN. Tafazoli et al (2017) probed the rat ventral pathway using a battery of isolated visual objects (rendered in white against a black background) undergoing a range of transformations (position, size, orientation, etc.). This was a rather complex image set for a rodent neurophysiology study - complex enough to reveal the increase of view-invariant object information shown in Fig. 1C. However, this stimulus set is much smaller and less complex than the Imagenet set used to train and test CNNs, including the pretrained VGG-16 network used in our study.
>
> In Figure 5, we decided to use Imagenet to probe the category information available in single units of the network. This is because the network was trained with Imagenet in the first place, but also because we believed that probing the network with such a natural image set would have made our conclusions more general and more interesting for the AI community. As a result, consistently with what already shown by Doimo et al (2020), the increase of category information emerged only in the last layers of the network. However, because of the differences with the stimulus set used by Tafazoli et al (2017), the evolution of category information observed across rat visual cortex (Fig. 1C) and VGG-16 (Fig. 5) can be compared only at the level of the global trends, not at the fine-grained level of an area-to-layer match. What is interesting in the comparison of Fig. 1C and 5 is that, for both systems, we see an accumulation of object information in individual units that parallel the capacity of populations of units in the various layers to support object classification.
>
> ### **2. Significance**
>
> > *The results are not particularly surprising …The pretrained-vs-random results are all unsurprising…*
>
> We do not agree that “the pretrained-vs-random results are all unsurprising”. As mentioned in the response to 2Y7t, our results show that information about several low-to-middle level properties, which are thought to be important for image processing in both biological and machine vision, is first accumulated and then discarded as a result of training a CNN for image classification. The strength of this suppression is such that, after training, individual units in the final convolutional layers retain less information about these properties than the corresponding units in a random network. This result has two important conceptual and practical implications. If the goal is to decode the orientation or contrast content of an image patch from a single unit, one will do a better job at reading out the activity at an initial layer of the network, rather than its output layers. Even more surprisingly, one will do a better job at reading out the activity of the final convolutional layers of an untrained, fully random network than those of the trained one. To our knowledge, **these conclusions cannot be inferred from previous work comparing visual cortex and CNNs.** Moreover, Fig. 4 shows how learning, at least for luminance and contrast, actively discards information about these features at every layer of the trained network. This suggests that accumulating information about certain visual properties, but then drastically pruning it, is a fundamental feature of feedforward CNNs trained for image classification. Again, **we are not aware of previous studies that have clearly and systematically shown this.**

---

> ### Author Response · Authors · 2022-08-01
> **Reply to reviewer sQm2 (Part I)**
>
> We thank the reviewer for their appreciation of the difference between our study and previous comparisons between visual cortex and CNNs. In our response, we will try to clarify further the originality of our approach and conclusions. We will respond to each point separately below.
>
> ### **1. Consistency between the trend found in visual cortex and the CNN**
>
> > *… the trend consistency between the visual cortex and the CNN is a bit overstated. The trends are only consistent if we consider a subset of layers …*
>
> In general, we agree with the reviewer that the similarity between artificial and natural neural networks is in terms of qualitative trends. **This is unavoidable, because, while for the CNN we have access to the entire hierarchy, for the rat visual system we only have access to a subset of processing stages.** This is a common issue to virtually every modeling study of visual cortex using CNNs. In the case of our study, we track the information flow from primary visual cortex (V1) throughout a progression of lateral visual cortical areas (LM, LI and LL) that, according to previous studies, are part of the rat object-processing pathway, but do not fully encompass it. Specifically, the areas that are missing from our analysis are those at the “front end” of the visual system: the retina (with its various layers, from photoreceptors to ganglion cells) and the thalamus. Additionally, our analysis does not consider TO, a downstream area from LL and, more importantly, all those high-level cortical areas at the “back end” of the object processing pathway - i.e., regions, such as perirhinal, posterior parietal (PPC) and prefrontal cortex (PFC), which are involved in memory, classification and decision making. Finally, compared to the primate ventral stream, the rat visual hierarchy is shallower (likely missing the different stages of the inferotemporal cortex). **Thus, with respect to the layers of a CNN, this would be equivalent to miss the first few convolutional layers (retina + thalamus), as well as some of the deepest convolutional layers (TO + the missing inferotemporal stages) and the fully connected ones (the classification/decision regions).** More specifically, a reasonable approach that works well in our data is to drop the first 3-4 layers, plus the last 3-4 convolutional layers and the fully connected ones. **_Rat visual areas V1, LM, LI and LL can be then roughly equated to layers 5-8 in VGG-16._** We will explain this in our revision for the camera-ready version.
>
> With these considerations in mind, we can address the individual concerns of the reviewer:
>
> > - *figure 1A vs figure 2 … The consistency assumes we ignore (roughly) layers 9-16.*
>
> This is correct and the reason is that, in the rat, we are missing at least one higher-order visual area after LL (i.e., TO), as well as the highest-level decision and categorization regions. The latter would correspond to the final layers of the CNN, where, according to Ansuini et al (2019), the intrinsic dimension becomes very low. Consistently, previous studies of PPC and PFC (already cited in our manuscript) have shown very low-dimensional representations of sensory stimuli. Thus, what we see, in the transition from LI to LL, is the beginning of the drop of dimensionality of the representation, after the increase along the initial processing stages.
>
> > - *figure 1B vs figure 3A/B. The decreasing trends are consistent, but only if we ignore layers 1-3.*
>
> For luminosity, a monotonic decreasing trend is observed in both cortex and the CNN. For contrast, in the cortex we miss the initial rise of the information that, in the CNN, takes place from layer 1 to 3. This suggests that contrast information, in the cortex, is concentrated in single neurons via processing taking place between the retina and V1 - **a conclusion that is fully consistent with the known properties of the retina**, where center-surround contrast detectors first emerge in the ganglion layer.
>
> > - *figure 1D vs figure 3C/D. Here the trends are not as consistent as stated…*
>
> As for contrast, in the cortex we miss the initial rise of orientation information, again because we miss the areas before V1. But we know that a rise must be taking place, because it is in V1 that orientation tuning emerges, as shown in many neurophysiological studies across species. This corroborates the intuition that rat V1 is roughly equivalent to layer 5 of VGG-16.
>
> Regarding corner information, this is the only property for which we cannot find a proper alignment between cortex and VGG, since it increases from V1 to LI, while it decreases from layer 5 to layer 6 of VGG, having already peaked in layer 5. On the other hand, the final part of the trend is consistent between the two systems, as corner information is stable (and quite large) from LI to LL and from layer 6 to layer 8.

---

### Official Review · Reviewer_2Y7t · 2022-07-08

**Rating:** 4
**Confidence:** 4
**Soundness:** 2 fair
**Presentation:** 3 good
**Contribution:** 1 poor

**Summary:**

The paper compares internal representations of what could be called the rat's ventral visual stream and the VGG-16 network trained on ImageNet. The authors find that the intrinsic dimensionality (ID) of rat ventral stream and VGG-16 follow a similar pattern, with ID first increasing and then decreasing again. In addition, they estimate the mutual information between some low-level image features and internal representations and also find similar trends.


**Questions:**

 1. While I find it quite interesting that the ID follows a similar pattern in both brain and ANN, several questions remain:

    - How general is this finding across different ANN architectures (e.g. ResNet, ViT)

    - The peak seems to be much earlier in VGG/AlexNet than in the brain. How do you explain this result?

 1. How can the mutual information increase over layers (Figs. 3–5)? Information cannot be created by processing (data processing inequality), so all information that is there about the stimulus in, say, layer 5 also has to be there already in layer 3. Thus, I believe that the fact that the MI curves increase in the first couple of layers in Figs. 3–5 is probably just an artifact of estimating MI from finite data. This makes me wonder how informative these curves are in general.

 1. Assuming the analyses were meaningful, it is not clear to what extent they represent general properties of artificial neural networks trained on ImageNet. Would these findings hold across a variety of more modern architectures? Do they depend on the training objective? Many questions remain open, and just analyzing VGG-16 will not be able to answer them.

 1. Suppose the representational similarities indeed existed, how would this result change how we think about or advance machine learning – the goal of this conference – or neuroscience? Wouldn't we want to know what causes these similarities (network architecture, training objective, dataset, etc.) or what we have to change things to destroy them?


**Limitations:**

Yes

**Strengths And Weaknesses:**

### Strengths
 + Understanding similarities and differences of real and artificial neural networks is an important topic
 + Paper is clearly written and easy to follow
 + Finding that ID follows a similar pattern in brain & ANN might potentially be interesting

### Weaknesses
 - Analysis way to restrictive, considering only one (fairly old) CNN architecture
 - Increasing MI values in Fig. 3–5 are implausible (-> data processing inequality)
 - Unclear how the analysis advances either ML or neuroscience

---

> ### Author Response · Authors · 2022-08-01
> **Reference List for reviewer 2Y7t**
>
> **Selected references for usage of VGG in the field of Visual Neuroscience**
>
> 1. Cadena, S et al. “How Well Do Deep Neural Networks Trained on Object Recognition Characterize the Mouse Visual System?” NeurIPS, 2019.
> 2. Dobs, K et al. “How Face Perception Unfolds over Time.” Nature Communications 10, no. 1 (March 19, 2019): 1258.
> 3. Geirhos, R et al. “Of Human Observers and Deep Neural Networks: A Detailed Psychophysical Comparison.” Journal of Vision 17, no. 10 (August 31, 2017): 806.
> 4. Grossman, S et al. “Convergent Evolution of Face Spaces across Human Face-Selective Neuronal Groups and Deep Convolutional Networks.” Nature Communications 10, no. 1 (October 30, 2019): 4934.
> 5. Güçlü, U et al. “Deep Neural Networks Reveal a Gradient in the Complexity of Neural Representations across the Ventral Stream.” Journal of Neuroscience 35, no. 27 (July 8, 2015): 10005–14.
> 6. Higgins, I et al. “Unsupervised Deep Learning Identifies Semantic Disentanglement in Single Inferotemporal Neurons.” Nature Communications 12, no. 1 (December 2021): 6456.
> 7. Jaegle, A et al. “Population Response Magnitude Variation in Inferotemporal Cortex Predicts Image Memorability.”. ELife 8 (August 29, 2019): e47596.
> 8. Jozwik, K et al. “Large-Scale Hyperparameter Search for Predicting Human Brain Responses in the Algonauts Challenge.” bioRxiv, August 14, 2019.
> 9. Kalfas, I et al. “Representations of Regular and Irregular Shapes by Deep Convolutional Neural Networks, Monkey Inferotemporal Neurons and Human Judgments.” PLOS Computational Biology 14, no. 10 (October 26, 2018): e1006557.
> 10. Kindel, W et al. “Using Deep Learning to Probe the Neural Code for Images in Primary Visual Cortex.” Journal of Vision 19, no. 4 (April 26, 2019): 29.
> 11. Luo, X et al. “The Costs and Benefits of Goal-Directed Attention in Deep Convolutional Neural Networks.” Computational Brain & Behavior 4, no. 2 (June 1, 2021): 213–30.
> 12. Mehrer, J et al. “An Ecologically Motivated Image Dataset for Deep Learning Yields Better Models of Human Vision.” Proceedings of the National Academy of Sciences 118, no. 8 (February 23, 2021): e2011417118.
> 13. Rust, N et al. “Understanding Image Memorability.” Trends in Cognitive Sciences 24, no. 7 (July 1, 2020): 557–68.
> 14. Schrimpf, M. et al. “Brain-Score: Which Artificial Neural Network for Object Recognition Is Most Brain-Like?” bioRxiv, January 2, 2020.
> 15. Seeliger, K. et al. “Convolutional Neural Network-Based Encoding and Decoding of Visual Object Recognition in Space and Time.” NeuroImage, 180 (October 15, 2018): 253–66.
> 16. Tripp, Bryan P. “Similarities and Differences between Stimulus Tuning in the Inferotemporal Visual Cortex and Convolutional Networks.” IJCNN, 3551–60, 2017.
> 17. Vinken, K. & Beeck, H. O. de. Using deep neural networks to evaluate object vision tasks in rats. PLOS Computational Biology 17, e1008714 (2021).
> 18. Zeman, A et al.  “Orthogonal Representations of Object Shape and Category in Deep Convolutional Neural Networks and Human Visual Cortex.” Scientific Reports 10, no. 1 (February 12, 2020): 2453.

---

> ### Author Response · Authors · 2022-08-01
> **Reply to reviewer 2Y7t (Part II)**
>
> ### **4. Our contribution**
> > *Suppose the representational similarities indeed existed, how would this result change how we think about or advance machine learning … or neuroscience?*
>
> Our results show that information about certain low-to-middle level properties, which are thought to be important for image processing in both biological and machine vision, is first accumulated and then discarded, as a result of training a CNN for image classification. The strength of this suppression is such that, after training, individual units in the final convolutional layers retain less information about these properties than the corresponding units in a random network. This result has two important conceptual and practical implications. If the goal is to decode the orientation or contrast content of an image patch from a single unit, one will do a better job at reading out the activity of the initial layers, rather than the output layers. Even more surprisingly, one will do a better job at reading out the activity of the final convolutional layers of an untrained, fully random network than those of the trained one. To our knowledge, these conclusions are far from trivial and original. Moreover, Fig. 4 shows how learning, at least for luminance and contrast, actively discards information about these features at every layer of the trained network. This suggests that accumulating information about certain visual properties, but then drastically pruning it, is a fundamental feature of feedforward CNNs trained for image classification. In this sense, we believe that our results will have an impact on how we think about hierarchical image processing, either biological or artificial.
>
> > *Wouldn't we want to know what causes these similarities …?*
>
> Indeed, it would be great to reach an understanding of what causes these similarities. For instance, understanding the origin of the characteristic profile of intrinsic dimensionality, and its potential connection to pruning of low-level visual information, could suggest new normative or algorithmic principles underlying the capacity of CNNs or the ventral stream to process visual data. However, **establishing that these similarities exist in the first place is a prerequisite for understanding why they exist.** This is the main contribution of our paper, in a long tradition of observational studies of neural systems: by highlighting the existence of common patterns between biological and artificial systems, we open the way to their further study.

---

> ### Author Response · Authors · 2022-08-01
> **Reply to reviewer 2Y7t (Part I)**
>
> We thank the reviewer for appreciating the general goal of our study, the ID findings and the presentation of our paper, and for organizing their more critical comments in a very clear way. We will respond to each point separately below.
>
> ### **1. Generality of the ID findings and comparison with the brain**
> > - *How general is this finding across different ANN architectures (e.g. ResNet, ViT)*
>
> The hunchback profile of the ID curve has been established as a general and robust property across multiple architectural families, including ResNets, in the existing literature (Ansuini et al 2019). We’ll make sure to highlight this more in the camera-ready version of the paper.
>
> > - *The peak seems to be much earlier in VGG/AlexNet than in the brain*
>
> This is similar to the general point made by reviewer sQm2 about the alignment of specific CNN layers with cortical areas. As discussed in our reply to that point, the cortical areas available in the rat recordings are only a subset of the full sensation-to-decision making pipeline represented by the artificial network. In particular, the earlier peak in VGG can be explained by considering that the last layers of the ANN would correspond to higher-order regions associated with memory, classification and decision making. These areas, which were not sampled in this set of cortical recordings, are known to contain very low-dimensional representations of sensory stimuli (see the references cited in our manuscript). Thus, what we see in Fig. 2B, in the transition from LI to LL, is the beginning of the drop of dimensionality of the representation, after the increase along the initial processing stages.
>
>
> ### **2. MI increase and data processing inequality**
> > *How can the mutual information increase over layers (Figs. 3–5)? Information cannot be created by processing (data processing inequality), so all information that is there about the stimulus in, say, layer 5 also has to be there already in layer 3*
>
> We thank the reviewer for bringing this up, giving us an opportunity to clarify a crucial point in our paper. **The MI values we report in our plots are not the MI between a stimulus feature and the activations of a layer taken collectively (across all the units)**, in which case the objection of the reviewer would be valid. **Ours are single-unit information estimates**, averaged over a sample of units in a given layer. **Therefore, the data processing inequality does not apply to (and is not violated by) our results.** For instance, if information about (say) orientation is distributed synergistically across multiple neurons in layer 3, but converges more onto individual neurons in layer 4, then the single-neuron orientation information will be higher in layer 4 than in layer 3 (this is exactly what happens, in visual cortex, in the transition from thalamus to V1). Of course, this doesn’t change the fact that the total amount of orientation information present in all layer 4 neurons taken collectively will be equal or lower than for layer 3. We will make sure this important point is made clearer in the final text.
>
>
> ### **3. Generality of MI findings**
> > *Assuming the analyses were meaningful, it is not clear to what extent they represent general properties of artificial neural networks ... Would these findings hold across a variety of more modern architectures ...?*
>
> The degree to which our results extend to different ANN architectures is a very interesting question. We focused on VGG because it is one of the most popular and influential CNN architecture families in visual neuroscience, commonly used either as a model of the ventral stream or as a benchmark against which such models should be tested (see below for a selection of relevant papers). Another important property of VGG, as of any other simple feedforward convolutional network, is that it allows for a very natural definition of the receptive field of individual units. This is fundamental for our analysis, because a notion of receptive field is required for the estimation of image feature information. By contrast, with more complex architectures, such as ResNets or ViTs, this concept would be harder to define, leading to the introduction of more hypotheses and more moving parts to the study.
>
> However, the point raised by the reviewer stands: showing the results in one network only seems overly restrictive. **We have therefore extended our analyses to other networks of the VGG family** (VGG-11 and VGG-19). In such networks, **we obtain very similar results to those in VGG-16.**
>
> This analysis is shown in a figure that is available at the following anonymous link: https://figshare.com/s/9a14ec02441a8f42cf08. We will add these results to the Appendix of the paper in its final version, and we will include a discussion of this point in a separate Limitations paragraph.

---

> > ### Comment · Reviewer_2Y7t · 2022-08-08
> > **Thanks for the clarifications**
> >
> > Thank you for clarifying. I'm happy to learn that the MI issue was a misunderstanding on my part!
> >
> > Regarding the generality, I'm somewhat disappointed by the response. It seems quite obvious that VGG-11 and 19 are not too different. Why not run the analysis on a different, more modern architecture like ResNet or ViT to show generality? This limited analysis undermines your point about the contributions. If your main contribution is to establish a similarity between artificial neural nets and the brain (rather than VGG and the brain), then I would have expected a more thorough investigation of whether these similarities are indeed properties of ANNs in general or those of a particular class of ANNs (VGG).
> >
> > Independent of this remaining concern, I'm increasing my score to reflect the fact that I had originally misunderstood the MI analysis in an important way.

---

### Official Review · Reviewer_vMFS · 2022-07-10

**Rating:** 8
**Confidence:** 4
**Soundness:** 4 excellent
**Presentation:** 4 excellent
**Contribution:** 4 excellent

**Summary:**

In this work, the authors have studied the similarity in the emergence of certain statistical patterns across neural activity recorded from the rat ventral visual stream and a (trained / randomly initialized) deep convolutional network optimized to perform large-scale object recognition on ImageNet. The authors present 3 interesting observations: 1) similar trend of intrinsic dimensionality of object representations across the rat ventral stream and pretrained deep convolutional networks 2) observed the pruning of low-level and mid-level information which followed their distillation in early layers of the rat ventral visual stream and VGG-16, these effects being a result of training and not merely a byproduct of the hierarchical processing architecture, 3) object categorization decoding accuracy closely tracked the increase of object-specific information across the VGG-16 hierarchy.


**Questions:**

- In Fig. 5, why do the authors think there is a sharp decrease in the decoding accuracy of the last two fully connected layers of the randomly initialized network, compared to the rest of the conv and fully connected layers that produce significantly better decoding of object category?

**Limitations:**

- As mentioned in the weaknesses, I would like the authors to please discuss limitations and provide directions for extending the current work.

**Strengths And Weaknesses:**

Strengths:
- There has been an explosion of recent studies exploring the similarity between biological and machine visual perception via representation similarity analysis and direct comparison of neural representations on a common stimulus set. This work on the other hand presents a less commonly studied yet important complementary direction of observing commonly emergent statistical trends in the representations of biological and artificial neural networks and highlights the change in these trends as a function of training.
- While the current study is restricted to comparing object recognition trends of deep networks with the rat ventral stream, the general methodology presented can be extended to primate analyses once the technology to record primate data across the ventral visual stream for a considerably large set of images is available.
- The authors provide adequate mathematical background in the Methods section to clearly follow the key results presented in the subsequent sections in comparing the statistical trends between rat ventral stream and VGG-16 representations.
Transparent disclosure of which neurons were discarded from the analyses and the criteria for discarding them (poor selectivity to the stimulus features being analyzed) in section 2.3.
- I found the paper to be very well-written and organized with clearly labeled plots (for the most part) along with detailed captions describing the key results are very helpful.


Weaknesses
- While the current analyses are interesting and sufficient to convey the key observations made by the authors, a high-level comment would be to relate the observed statistical trends with those obtained by the RSA studies measuring brain-CNN similarity. For e.g. it appears that a related finding is shown in https://arxiv.org/abs/1807.00053 in Fig 5, wherein there exists a correspondence between the decoding accuracy of successive layers of a pre-trained deep neural network with successive areas of the primate ventral visual stream.
- Nit: Some of the plots may be hard to read due to the minor difference in the color or linestyle of the various markers. E.g. the authors must please check if Fig 4’s coloring scheme is accessible to color-blind readers. It was difficult for me to initially identify the solid vs dashed lines in Fig. 5 as they look quite similar here. Similar to Fig. 3, I would add plot titles “orientation” and “contrast” to Fig 4.A and 4.B for improved readability.
- The authors could potentially discuss more the limitations of the current work and highlight areas of improvement/extension that could be useful to inspire future work.

---

> ### Author Response · Authors · 2022-08-01
> **Reply to reviewer vMFS**
>
> We thank the reviewer for the useful feedback. In answer to their question, the sharp decrease in the decoding accuracy in the fully connected layers of the random network may simply be a dimensionality effect. In fact, the dimensionality of the neural spaces decreases from 4096 neurons to 1000 neurons when going from layer 14 to layer 16 in the network. Additionally, there could be an increased averaging effect, as neurons in the fully connected layers all share the same inputs, unlike those in the convolutional layers.
>
> In answer to the other points raised under “weaknesses”:
> - We thank the reviewer for stressing the connection with existing brain-CNN similarity studies. We already discuss some of those in our paper (see also our reply to reviewer DRfj), but we will make sure to include the RSA angle.
> - We thank the reviewer for catching the issue with the colors in the plots. They will be fixed in the camera ready version.
> - We will include a separate limitations paragraph in the camera ready version (see also responses to reviewers 2Y7t and sQm2).

---

> > ### Comment · Reviewer_vMFS · 2022-08-09
> > **Thanks for the response!**
> >
> > I thank the authors for their response to my questions and comments. I stick to my earlier opinion of the paper presenting a very interesting and underexplored direction to compare emergent statistical trends across artificial and biological representations. I think this work will be of great interest to the computational neuroscience community at NeurIPS.

---

### Official Review · Reviewer_DRfj · 2022-07-11

**Rating:** 6
**Confidence:** 3
**Soundness:** 3 good
**Presentation:** 3 good
**Contribution:** 3 good

**Summary:**

The paper analyzes whether low- and mid-level feature information is clearly represented in early CNN layers, but not as clearly represented in later CNN layers, a property that is present in early and late vision processing systems in mice. In particular, the low- and mid-level features of luminosity, contrast, orientation, and corners are investigated. Additionally, the paper shows that the intrinsic dimension of representations is relatively low in early CNN layers, but much higher in later CNN layers.

**Questions:**

One thing to note is that prune and distill have defined meanings in machine learning model compression (Graves, 2011; Hinton et al., 2014; Blundell et al., 2015), so their use in this paper could be jarring to some readers.

Cadena et al. (2019) found that ImageNet trained CNNs are not a particularly good model of the mouse visual cortex. Discussing how their results relate to your approach in the introduction would be good.

How does your work on the dimensionality of different neural network layers add to the work presented in Güçlü and van Gerven (2015), who show that the Kolmogorov complexity of layer representations increases with depth?

Why look at the pre-activations when analyzing the activations of a neural network layer? Isn't the information accessible by the next layer dependent on what is let through by the activation function?

Graves, A. (2011). Practical variational inference for neural networks. Advances in neural information processing systems, 24.

Hinton, G., Vinyals, O., & Dean, J. (2014). Distilling knowledge in a neural network. NIPS 2014 Deep Learning Workshop.

Blundell, C., Cornebise, J., Kavukcuoglu, K., & Wierstra, D. (2015, June). Weight uncertainty in neural network. In International conference on machine learning (pp. 1613-1622). PMLR.

Cadena et al. (2019). How well do deep neural networks trained on object recognition characterize the mouse visual system? NeurIPS 2019 NeuroAI Workshop

Güçlü, U., & van Gerven, M. A. (2015). Deep neural networks reveal a gradient in the complexity of neural representations across the ventral stream. Journal of Neuroscience, 35(27), 10005-10014.


**Limitations:**

The authors should add a discussion on the limitations of their work to the Conclusions and Discussion section. One limitation that potentially could be discussed is that high level statistical properties can be shared by systems that process information differently.

**Strengths And Weaknesses:**

Overall, the paper is reasonably written and effectively conveys what the authors did. However, the originality and significance of the paper are, in my opinion, not very high. While specifically looking at luminosity, contrast, orientation, and corner information is nice, the idea that early feature information is prominent in early CNN layers and is lost in later layers is not new. Nor is the idea that the complexity of layer representations increases from early to late CNN layers. One extremely relevant paper on the topic is Güçlü and van Gerven (2015), which showed that information present in early human visual areas is prominent in early CNN layers, but not in later CNN layers. That paper also showed that the Kolmogorov complexity of layer representations increases with depth. The analysis of mutual information between layer representations and labels is not new, either [35]. The comparison of CNNs to mouse data is, also, not new (Cadena et al., 2019). Cadena et al. (2019) found that CNNs were a poor model of the visual system of mice. If the submitted paper could address, and present evidence for, why their results contradict the findings of Cadena et al. (2019), it would strengthen the paper.

---

> ### Author Response · Authors · 2022-08-01
> **Reply to Reviewer DRfj (Part II)**
>
> ### **3. Novelty of the mutual information analysis (2017)**
> > *The analysis of mutual information between layer representations and labels is not new, either [35].*
>
> The idea of analyzing information between labels and activations of a deep neural net is not one of our claims to novelty. Clearly, mutual information is a popular metric and its use is ubiquitous in deep learning and neuroscience. What is novel in our approach is the systematic analysis of the information about several meaningful and well-defined visual features carried by units at different stages of artificial and biological neural systems for vision, and its comparison to the intrinsic dimensionality of neural representations. The fact that this analysis is performed using mutual information is a simple consequence of the usefulness and generality of this metric, which allows us to put all the features of interest on the same footing. The image label is just one of the features we’re interested in, but does not possess any special significance, besides being important for the purpose of the studied systems. By contrast, in ref [35] the use of mutual information - and not some other measure - is key to the theoretical point being developed, which is based on the Information Bottleneck method. And similarly for the choice of analyzing the data labels: labels are special in [35], because it is what the network is trying to predict, and labels are the only feature the authors can investigate, given that their small, fully-connected network operates on synthetic binary inputs, not on actual images.
>
> ### **4. Relationship with Cadena et al 2019**
> > *The comparison of CNNs to mouse data is, also, not new (Cadena et al., 2019)...*
>
> With respect to the work of Cadena et al. (2019), not only our study is different because of our different approach and goals (same arguments used in discussing the differences with Güçlü and van Gerven, 2015), but, more substantially, because our analyses are based on data taken from the **_rat_ visual system** and **not from the _mouse_ visual system** (as in Cadena et al).
>
> In the rat, the existence of an object-processing pathway (areas V1, LM, LI and LL) has been established in a series of studies showing a clear gradient of ventral stream properties from V1 to LL (see Fig. 1C-D). This is why we could hope to find similar functional trends in a CNN: because they existed in the first place in the rat. However, and crucially for the reviewer’s point, no similar trends have been reported to date in the mouse. As a matter of fact, evidence for the existence of an object-processing pathway is still very limited in the mouse, so it is not surprising that image classification machines as deep CNNs are poor models of the mouse visual system. This explains at once why our study is novel compared to Cadena et al. (2019), and why our results are not in conflict with their findings. We will discuss this in the revised manuscript.
>
> ### **5. Pre vs post nonlinearity analysis**
> > *Why look at the pre-activations …? Isn't the information accessible by the next layer dependent on what is let through by the activation function?*
>
> In our view this choice is somewhat arbitrary: the post-relu activations do indeed represent the information transmitted to the next layer, but by the same token the linear activations represent the information received by one neuron from the previous layer. Both choices allow measuring information and comparing it between layers in a consistent way. One possible difference is that, because the activation function simply maps half of the possible values to zero, this could make it harder to spot interesting patterns in the information, decreasing the range over which this can vary between layers. In practice, we have now checked that **if one performs our analysis using the (post-relu) activations, the results are qualitatively similar**, with the exception of the decreasing trend of the corner information which seems to be not discernible anymore amidst the layer-to-layer fluctuations.
>
> This analysis is shown in a figure that is available at the following anonymous link: https://figshare.com/s/dacf8733b6d310896c4b. We will add this figure to the Appendix of our camera ready revision.
>
> ### **4. Prune, distill**
>
> We have chosen the words prune and distill because they hint to the main result of our paper, namely the fact that low-level features are pruned from the images aggressively before distilling the high-level features. However, we understand that the words "prune" and "distill" can be misleading for computer scientists. If the panel deems it appropriate, we would be happy to drop the first part of the title, changing it to "Similar reformatting of image information along rat visual cortex and deep neural networks"
>
> **References**
>
> Baker, Ben, Benjamin Lansdell, and Konrad Kording. “A Philosophical Understanding of Representation for Neuroscience.” arXiv, April 28, 2021.

---

> > ### Comment · Reviewer_DRfj · 2022-08-02
> > **Reply to Author-specific Rebuttal**
> >
> > Thank you for the clear and detailed responses to my questions.
> >
> > 1. Thank you for pointing out the differences between how Güçlü and van Gerven computed the Kolmogorov complexity of a layer's representation and how you computed the intrinsic dimension (ID) of a layer's representation. The presence of the "hunchback" profile in the ID plots does distinguish the results of the two approaches, and does, in my perspective, increase the novelty of your paper.
> >
> > 2. The point is taken that the cited literature does not explicitly demonstrate specific low-level feature information that is lost in later layers. These results are not surprising, but do add to the literature.
> >
> > 3. The point is taken that the mutual information analysis is not one of the main novelty claims of the paper and is present to add another measure for the phenomenon being studied.
> >
> > 4. Thank you for pointing out the difference between the visual system of mice and rats and pointing out that your paper is dealing with the rat visual system. I am sorry for making that mistake.
> >
> > 5. Thank you for pointing out the difference between what is being measured when looking at pre-activation and activation representations and for performing the additional analyses using the activations.
> >
> > 6. Thank you for your willingness to rephrase the title of the paper to be less confusing to some computer science audiences. I realize that this change may not be made, but the willingness is nice, regardless.
> >
> > As a result of these clarifications, I have increased my contribution score from 1 to 3 and my rating from 4 to 6.

---

> ### Author Response · Authors · 2022-08-01
> **Reply to Reviwer DRfj (Part I)**
>
> The main concern of the reviewer is the novelty and originality of our results, as compared to previous studies that have used CNNs to model and understand visual cortex. We believe that our approach is different from this previous work and allows understanding important properties of feature processing in both cortex and CNNs in a way that previous modeling studies do not. More specifically, as a reply to the reviewer’s questions:
>
> ### **1. Kolmogorov complexity in Güçlü and van Gerven (2015) vs intrinsic dimensionality**
> The concept of representation, and the way its complexity is measured by Güçlü and van Gerven, are fully unrelated to what we measure with the intrinsic dimensionality of neural activity.
>
> In Güçlü and van Gerven, the representation of a neuron is defined as the deconvolution of the activity of that neuron, conditional on a particular image from the stimulus set that maximizes neural activation. So, the “representation” of one neuron is one image, corresponding to a set of features that strongly activate that neuron (a proxy for the “filter” that the neuron applies to input images). The complexity of that image is its compressed size, as a file on disk, measured using an arbitrary binary encoding and compression scheme (this is meant as an approximation of Kolmogorov complexity). The representation complexity of a layer is the average complexity of the units in that layer. In this sense, the representation complexity in Güçlü and van Gerven measures the complexity of the image features that activate the neurons in a certain layer, computed separately for each neuron and averaged across the layer.
>
> In contrast, intrinsic dimensionality (ID) is a property of how the response of a neural population (not a single unit) varies as the input varies. It is the minimal number of coordinates which are necessary to describe a set of neural population vectors, each corresponding to the activity elicited by a distinct visual stimulus, without significant information loss. It cannot be computed for a single neuron, and importantly it does not only depend on neurons’ maximal responses, but more generally on their responses across a stimulus set (i.e., the response manifold).
>
> **Conceptually, then, the two metrics are unrelated.** Quantitatively, they behave differently (representation complexity as in Güçlü and van Gerven grows monotonically with depth, while ID does not, exhibiting the hunchback profile reported in fig 1A and 1B), but this is not even a contradiction - just two answers to different questions.
>
> Finally, please note that throughout our text we use the word “representation” very differently than in Güçlü and van Gerven. In our paper, given a stimulus, we call neural representation of that stimulus the pattern of neural activation elicited by that stimulus across a population of either cortical or artificial units. In Güçlü and van Gerven, given a neuron, a representation is a stimulus pattern that evokes a strong response in that neuron. Although the word “representation” is notoriously overloaded with different meanings in neuroscience, we believe our usage conforms to the one that will be least surprising for the majority of readers [Baker, Landsell and Kording 2021].
>
> ### **2. Novelty with respect to the work by Güçlü and van Gerven (2015)**
>
> > *… the idea that early feature information is prominent in early CNN layers and is lost in later layers is not new. Nor is the idea that the complexity of layer representations increases from early to late CNN layers.*
>
> In Güçlü and van Gerven, the authors show that: 1) voxel activity in deep visual cortical areas is better predicted by deeper than earlier CNN layers; and 2) the preferred stimuli of deeper layers are more complex than those of earlier layers. Although these findings are suggestive of a loss of low-level feature information along the CNN, they do not directly demonstrate it, neither they quantify its magnitude.
>
> **More importantly, our study goes well beyond reporting a loss of early feature information along the processing hierarchy.** Our study actually shows that, for several properties, information does not simply monotonically decay - it increases and concentrates first along the few initial layers and only later gets pruned (as shown in Fig. 3). Our study also shows that such a pruning is actively instantiated and repeated across every layer of the hierarchy (as shown in Fig. 4), to the extent that information about some image features is so drastically reduced in deep layers to be actually larger for random networks. None of these conclusions (1. the non-monotonic trends of feature information; 2. the active pruning across the entire network; and 3. the comparison with untrained networks) can be found in, or trivially inferred from, Güçlü and van Gerven (2015).

---

### Meta-Review · Area_Chair_Qxc9 · 2022-08-23

**Recommendation:** Accept
**Confidence:** Less certain

**Metareview:**

This paper examines the intrinsic dimensionality (ID) of representations in the rat brain and CNNs. The authors show that the rat brain, like the CNNs studied, have distinct expansion-contraction phases, and that in the CNNs one also observes a distillation and pruning of low- and mid-level information, similar to what is seen in the brain. The authors also show that in the CNNs high-level object information only emerges after these steps of distillation and pruning. This illustrates potentially interesting parallels between information processing in the real brain and CNNs.

The reviewers were split on this paper. The initial reviews identified issues of novelty and insight. But, after the author responses, three out of four agreed that the work was interesting and technically sound, and they found the paper well-written. One reviewer was still concerned that the paper does not do enough to show that these similarities in ID and information retention tell us anything meaningful. They were also concerned that comparisons were not made across enough architectures. Nonetheless, given the balance in the reviews and post response scores, an accept decision was reached.

**Award:**

No

---

### Decision · Program_Chairs · 2022-09-14

Accept